# CATMARK: A CONTEXT-AWARE THRESHOLDING FRAMEWORK FOR ROBUST CROSS-TASK WATERMARKING IN LARGE LANGUAGE MODELS

## ABSTRACT

Watermarking algorithms for Large Language Models (LLMs) effectively identify machine-generated content by embedding and detecting hidden statistical features in text. However, such embedding leads to a decline in text quality, especially in low-entropy scenarios where performance needs improvement. Existing methods that rely on entropy thresholds often require significant computational resources for tuning and demonstrate poor adaptability to unknown or cross-task generation scenarios. We propose Context-Aware Threshold watermarking (CATMARK), a novel framework that dynamically adjusts watermarking intensity based on real-time semantic context. CATMARK partitions text generation into semantic states using logits clustering, establishing context-aware entropy thresholds that preserve fidelity in structured content while embedding robust watermarks. Crucially, it requires no pre-defined thresholds or task-specific tuning. Experiments show CATMARK improves text quality in cross-tasks without sacrificing detection accuracy.

## 1 INTRODUCTION

The expanding capabilities of Large Language Models (LLMs) have enabled their application in increasingly diverse and sophisticated generation tasks Zhao et al. (2025), from acting as AI agents that produce structured data to solving complex scientific problems and writing functional code Chen et al. (2021); Guo et al. (2024). However, this proliferation of high-quality, machine-generated content poses formidable challenges for authenticity verification Burrus et al. (2024); Ayoobi et al. (2024) and the prevention of misuse Ayoobi et al. (2023); Dammu et al. (2024). Text watermarking, which embeds imperceptible statistical signals into generated text, has emerged as a promising solution for establishing content provenance Liu et al. (2024); Chen et al. (2023); Yoo et al. (2023). The dominant paradigm involves augmenting the model's output logits; a foundational method, for example, partitions the vocabulary into "green" and "red" lists and adds a positive bias to the logits of green-listed tokens to embed a detectable signature Kirchenbauer et al. (2023).

Initial research quickly identified a primary limitation of this approach: its performance degrades significantly in low-entropy contexts, such as code generation, where modifying deterministic tokens can corrupt functional correctness. To address this, subsequent work has focused on entropy-aware adaptations. SWEET Lee et al. (2023) introduced a static entropy threshold, selectively applying the watermark only to high-entropy tokens to preserve low-entropy syntactic structures. Building on this, EWD Lu et al. (2024) refined the detection process by assigning weights to tokens proportional to their entropy, improving sensitivity without a hard threshold. While these methods marked important progress for single-domain tasks, they addressed only part of the problem.

The primary remaining challenge, which we identify as the core of our work, is the absence of a robust watermarking solution for cross-task generation scenarios. Modern LLMs are increasingly deployed in complex workflows where they must seamlessly switch between different generation modalities within a single output sequence Shoshan et al. (2025). For instance, an AI agent may generate executable code (low entropy) interwoven with natural language documentation (high entropy), or a mathematical reasoning agent might produce structured formulas alongside explanatory text. Existing methods are ill-equipped for such heterogeneous outputs. A single, static entropy threshold, as used in SWEET, is fundamentally inadequate; a threshold calibrated for natural language will be too permissive for code, harming its correctness, while one set for code will be too restrictive for text, rendering the watermark undetectable. This forces a compromise that fails to satisfy the requirements of either task Liu & Bu (2024); Chen et al. (2023). Furthermore, detection schemes that treat the entire text uniformly, like EWD, cannot adapt to these sharp, context-driven shifts in entropy, diluting the statistical signal and weakening detectability.

To address this critical gap, we propose the **Context-Aware Threshold Watermark** (CATMARK), a novel framework that dynamically adapts its watermarking strategy to the local context of the generated text. Instead of relying on a single, global threshold, CATMARK employs a lightweight token categorization mechanism to identify the current generation context (e.g., code versus natural language) and computes a distinct, tailored entropy threshold for each. This allows it to selectively apply a strong watermark to high-entropy text while preserving the integrity of structured, low-entropy code, all within a single, continuous output. This adaptive approach eliminates the need for manual, task-specific tuning and ensures robust performance across diverse and mixed-modality generation tasks. Our contributions are threefold:

- **Cross-Task Robustness:** We are the first to systematically investigate and address the challenge of watermarking in cross-task generation scenarios. We introduce a quality-aware evaluation framework to rigorously assess performance in settings that mix modalities, such as code generation with inline documentation.

- **Dynamic Threshold Automation:** We introduce a novel dynamic thresholding mechanism that first categorizes tokens into context-specific clusters based on the KL divergence of their logit distributions from learned prototypes. It then automatically computes adaptive entropy thresholds using quantiles of the historical entropy distribution within each category, enabling real-time adaptation to varying textual complexities without manual intervention.

- **Theoretical and Empirical Validation:** We establish a theoretical lower bound for the detection z-score under our adaptive thresholding and provide extensive empirical evidence of its superiority. Our method significantly improves both output quality and detection robustness, achieving top-tier results such as a pass@1 score of 82.3% on HumanEval and a 100% AUROC on StackEval, simultaneously outperforming baseline methods across all cross-task benchmarks.

By solving the limitations of the static watermarking paradigm, CATMARK facilitates the practical and safe deployment of LLMs in the complex, multi-faceted applications where they are increasingly utilized, ensuring reliable content provenance without compromising functional integrity.

## 2 RELATED WORK

**Watermarking in Language Models.** Watermarking techniques aim to embed imperceptible signatures into model outputs for origin verification and misuse prevention Kirchenbauer et al. (2023); Hou et al. (2023). Red/green list-based methods modify sampling distributions to increase the frequency of selected tokens, achieving high detectability but often degrading generation quality Tu et al. (2023); Chang et al. (2024). Fixed-threshold strategies like KGWand SWEET Lee et al. (2023); Kirchenbauer et al. (2023) embed watermarks in tokens exceeding a preset entropy value, but are brittle in low-entropy settings such as code generation or structured data outputs Baldassini et al. (2024); He et al. (2024). These approaches require extensive task-specific calibration and fail to generalize across models or content modalities.

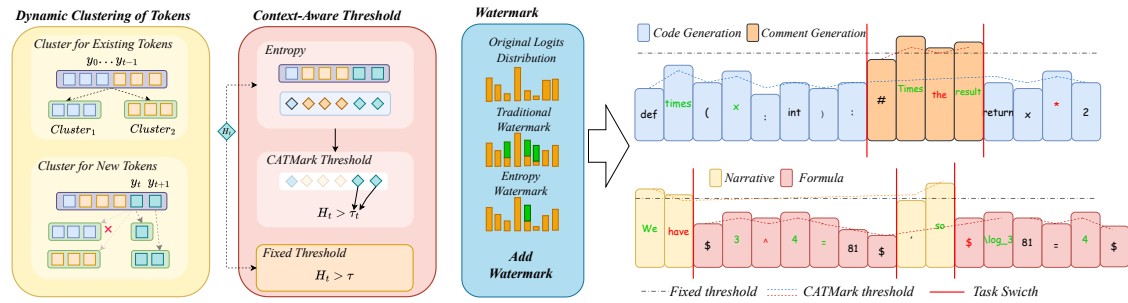

Figure 1: Comparison between static-threshold watermarking and our context-aware, cluster-based thresholding method, CATMARK. Our approach dynamically clusters generated tokens based on logit similarity (left panel), then computes a context-specific entropy threshold per cluster using historical entropy sequences (middle panel). Tokens whose entropy exceeds the adaptive threshold are watermarked (right panel). In the token sequence visualizations, rectangle height represents normalized entropy.

**Entropy-Adaptive and Low-Entropy Watermarking.** Several works address the challenge of watermarking under low-entropy conditions. STA-1 and STA-M Mao et al. (2024) introduce unbiased sampling and dynamic acceptance strategies, improving robustness without modifying logits, yet still depend on fixed green list proportions. Entropy-weighted detection methods (EWD) Lu et al. (2024); Räz (2024) enhance sensitivity by assigning entropy-proportional token weights at detection, but do not adapt watermark embedding during generation. Similarly, SWEET Lee et al. (2023) statically filters high-entropy tokens to preserve code correctness, though it lacks task-adaptive thresholding. While Liu & Bu (2024); Yoo et al. (2023) explore adaptive entropy-aware embedding, they either rely on external estimation modules or precomputed thresholds, which limit scalability.

**Cross-Task and Multimodal Generalization.** Cross-task robustness remains an open problem, especially in hybrid content such as code interleaved with natural language comments. Methods like POSTMARK Chang et al. (2024), RE-MARK-LLM Zhang et al. (2024), and VLPMarker embed watermarks without model access or via backdoor triggers, showing promise across tasks, but exhibit sensitivity to distribution shifts and entropy inconsistencies Christ et al. (2024); Nie & Lu (2024). Surveys by Liu et al. Liu et al. (2024) and Liang et al. Liang et al. (2024) highlight the shortcomings of static-threshold watermarking in dynamic and multimodal scenarios, especially in code generation tasks where entropy can fluctuate sharply across tokens Baldassini et al. (2024); Hu et al. (2023). Furthermore, multilingual and cross-lingual settings introduce semantic drift, making consistent watermark preservation harder Huang et al. (2023); Gloaguen et al. (2025).

To address these limitations, we propose Context-Aware Threshold Watermarking (CATMARK), a framework that dynamically adjusts the entropy threshold based on historical token entropy distributions. Unlike prior works relying on fixed or manually tuned thresholds Lee et al. (2023); Kirchenbauer et al. (2023), CATMARK leverages quantile-based entropy sampling to select watermark positions in real time, enhancing robustness across tasks and models. The weighted detection mechanism further amplifies signal strength in low-entropy contexts, ensuring watermark effectiveness without compromising text quality Liu & Bu (2024); Chang et al. (2024).

## 3 METHOD

We propose CATMARK, a context-aware watermarking framework that builds upon the foundation of statistical watermarking. Similar to established methods, its core principle is to embed a detectable signal by

subtly modifying the token sampling process. This is achieved by pseudorandomly partitioning the vocabulary $\mathcal{V}$ at each generation step $t$ into a "green list" ($\mathcal{G}_t$) and a "red list" ($\mathcal{R}_t$) based on a secret key and the preceding context. A positive bias, $\delta$, is then added to the logits of all tokens in $\mathcal{G}_t$, increasing their probability of being selected.

By selectively embedding watermarks only in high-entropy tokens within each semantic context, CATMARK achieves robust detectability while minimizing perturbation to structured content such as source code. Unlike static thresholding methods, our approach eliminates the need for manual tuning and adapts to varying content types within a single sequence.

### 3.1 GENERATION

The watermark generation process is outlined in Algorithm 1. For a tokenized prompt $\mathbf{x} = \{x_0, \ldots, x_{M-1}\}$ and a partially generated sequence $\mathbf{y}_{[:t]} = \{y_0, \ldots, y_{t-1}\}$, the model first computes the entropy $H_t$ of the next-token probability distribution.

A core innovation of CATMARK is the dynamic categorization of generation states. We maintain a set of active categories $\mathcal{C} = \{C_1, \ldots, C_K\}$, each defined by a prototype logits vector $\mathbf{p}_k \in \mathbb{R}^{|\mathcal{V}|}$. At step $t$, we compute a similarity score $d_k$ between the current logits vector $\mathbf{s}_t$ and each prototype $\mathbf{p}_k$ using the negative KL divergence:

$$d_k := -\text{KL}\left(\sigma(\mathbf{s}_t) \parallel \sigma(\mathbf{p}_k)\right) = \sum_{i=1}^{|\mathcal{V}|} \sigma(\mathbf{s}_t)_i \log \frac{\sigma(\mathbf{p}_k)_i}{\sigma(\mathbf{s}_t)_i}, \tag{1}$$

where $\sigma$ denotes the softmax function. The category $C_{k^*}$ with maximum similarity $d_{k^*}$ is selected. If $d_{k^*} \geq \alpha$ (where $\alpha$ is a similarity threshold), the token is assigned to $C_{k^*}$ and the prototype is updated via cumulative moving average:

$$\mathbf{p}_{k^*} \leftarrow \frac{N_{k^*}\mathbf{p}_{k^*} + \mathbf{s}_t}{N_{k^*} + 1}, \quad N_{k^*} \leftarrow N_{k^*} + 1, \tag{2}$$

where $N_{k^*}$ is the sample count for category $C_{k^*}$. Otherwise, a new category $C_{K+1}$ is initialized with $\mathbf{p}_{K+1} = \mathbf{s}_t$ and $N_{K+1} = 1$.

Once the active category $C_k$ is determined, its entropy history $H_{h,k}$ is used to compute the threshold $\tau_k$. Let $\rho$ represent a predefined minimum historical length. The threshold $\tau_k$ is calculated as:

$$\tau_k = \begin{cases} 0 & \text{if } |H_{h,k}| \leq \rho, \\ Q_{H_{h,k}}\left(f(\mu_{H_{h,k}})\right) & \text{otherwise,} \end{cases} \tag{3}$$

where $\mu_{H_{h,k}} = \frac{1}{|H_{h,k}|} \sum_{H \in H_{h,k}} H$ is the mean historical entropy for category $k$. When $|H_{h,k}| \leq \rho$, watermarks are applied unconditionally ($\tau_k = 0$). Otherwise, $\tau_k$ is set to the quantile of the entropy history corresponding to the cumulative probability $q = f(\mu_{H_{h,k}})$, i.e., the value satisfying:

$$\frac{1}{|H_{h,k}|}|\{H' \in H_{h,k} \mid H' \leq \tau_k\}| = q = f(\mu_{H_{h,k}}) \tag{4}$$

where $f$ is a function that maps the mean entropy to a cumulative probability value. In our implementation, we specifically choose $f(x) = e^{-x}$. The rationale for this choice and its empirical validation are discussed in Appendix F.

Finally, the vocabulary is partitioned into green and red lists with proportion $\gamma$. For tokens where $H_t > \tau_k$, a constant bias $\delta$ is added to the logits of green-listed tokens. Low-entropy tokens ($H_t \leq \tau_k$) are sampled without modification.

## 3.2 DETECTION

The detection algorithm is detailed in Algorithm 2. Since cluster assignments are unavailable at detection time, the process operates on the full sequence but uses entropy-based weighting to focus on regions where the watermark was most likely embedded.

Detection follows a statistical hypothesis testing approach. The null hypothesis ($H_0$) is that the text is natural and contains no watermark, meaning the number of green tokens should be statistically consistent with random chance.

Given a token sequence $y = \{y_0, \ldots, y_{N-1}\}$, the objective is to detect the presence of a watermark. Similar to the generation phase, the entropy $H_t$ is computed for each token $y_t$. The entropy sequence for all $N$ tokens is denoted as $H = \{H_0, \ldots, H_{N-1}\}$. The detection threshold $\tau$ is calculated as:

$$\tau = Q_H\left(f(\mu_H)\right), \tag{5}$$

where $\mu_H = \frac{1}{N} \sum_{i=0}^{N-1} H_i$ is the mean entropy of the sequence and $f$ is the function defined previously.

Inspired by EWD Lu et al. (2024), the influence of a token on the detection outcome is modeled as positively correlated with its entropy. For each token $y_t$ with an entropy value $H_t > \tau$, its weight $W_t$ is defined as a function of its entropy:

$$W_t = w(H_t), \tag{6}$$

where $w$ is a weighting function, which we set as $w(x) = x$.

The detection process proceeds as follows: First, the model's logits for each token are computed to obtain its entropy $H_i$. Next, a set of indices $\mathcal{I} = \{i \mid H_i > \tau\}$ is identified, corresponding to all tokens eligible for watermarking. For each token in this set, the green list $\mathcal{G}$ is reconstructed using the detection key and preceding tokens. Finally, the observed weighted sum of green tokens, $|s|_G$, is aggregated.

The $z$-score measures how far the observed sum of green token weights deviates from the expected sum under the null hypothesis. A high $z$-score indicates it is unlikely the text is natural, leading to the rejection of $H_0$ and detection of the watermark. The $z$-score is computed over the set $\mathcal{I}$:

$$z = \frac{|s|_G - \gamma \sum_{i \in \mathcal{I}} W_i}{\sqrt{\gamma(1-\gamma) \sum_{i \in \mathcal{I}} W_i^2}}, \tag{7}$$

where $|s|_G = \sum_{i \in \mathcal{I}, y_i \in \mathcal{G}} W_i$ is the observed weighted sum of green tokens. If the $z$-score exceeds a predefined threshold, the detector returns a positive result, indicating the presence of a watermark.

## 3.3 THEORETICAL ANALYSIS OF DETECTABILITY

CATMARK achieves a provably higher lower bound on the watermark detection $z$-score than the baseline method, EWD, thereby enhancing detectability. Theorem 1 formalizes this improvement. It demonstrates that by selectively excluding low entropy tokens which under specific conditions contribute negatively to the signal myalgo establishes a more robust statistical test. Our theoretical analysis employs spike entropy, a variant of entropy introduced by Kirchenbauer et al. (2023) to quantify this effect. The full proof is provided in Appendix H.

**Theorem 1.** *Given a token sequence $y = \{y_0, \ldots, y_{N-1}\}$ generated by a watermarked LLM, let $(S_0, \ldots, S_{N-1})$ be the corresponding sequence of spike entropies. If a token $y_j$ satisfies the low-entropy condition*

$$S_j < \gamma + (1-\gamma)e^{-\delta} \tag{8}$$

*then excluding this token from the $z$-score calculation, as is done in* CATMARK*, results in a higher lower bound on the $z$-score compared to including it, as in* EWD*. Here, $\gamma$ is the green-list ratio and $\delta$ is the positive logit bias.*

## 4 EXPERIMENTAL SETUP

This section presents a comprehensive experimental evaluation of our proposed watermarking technique for text generation. Our primary objectives are to assess (1) the preservation of output quality under watermarking and (2) the detectability of embedded watermarks. We conduct experiments using Qwen2.5-Coder-14B-Instruct Hui et al. (2024), a 14-billion-parameter instruction-tuned model optimized for code-related tasks, and Qwen2.5-14B-Instruct Team (2024) for mathematical and programming assistant tasks.

### 4.1 TASKS AND DATASETS.

Large Language Models (LLMs) are frequently deployed in cross-task settings; for instance, a code agent may be required to generate executable code, inline comments, and natural language explanations simultaneously. Watermarking may interfere with this multi-task generation capability. To evaluate such effects, we design two cross-task scenarios:

**Code Generation Task.** We evaluate on two widely used benchmarks: HumanEval Chen et al. (2021) and MBPP Austin et al. (2021). Both datasets contain Python programming problems, test cases, and human-written reference solutions. Models are asked to perform two tasks: generate code from a problem description and generate line-by-line comments for each generated code snippet. This dual requirement enables evaluation of both functional correctness and cross-task alignment between code and natural language.

**Question Answering Task.** We utilize the MATH-500 dataset Hendrycks et al. (2021), which requires models to parse a natural language problem, provide derivations and step-by-step reasoning about formulas based on the questions, and generate a final answer. This aims to evaluate the impact of watermarking on switching between structured text and logical narrative generation tasks. Additionally, to simulate real-world developer assistance scenarios, we employ the StackEvalbenchmark Shah et al. (2024). StackEval comprises 925 curated questions from Stack Overflow, spanning multiple programming languages and difficulty levels (Beginner, Intermediate, Advanced), covering code writing, debugging, code review, and conceptual understanding. We select the first 500 Intermediate-level questions to ensure both challenge and representativeness.

### 4.2 BASELINES AND EVALUATION METRICS.

For watermarking, we selected KGW Kirchenbauer et al. (2023), SWEET Lee et al. (2023), and EWD Lu et al. (2024) as baseline methods. These methods embed watermarks by distorting the model's sampling distribution. Although they have good detection performance, they also lead to a decrease in text quality. Among them, SWEET proposed to selectively embed and detect watermarks by setting a static threshold for a single task, while EWD introduced the detection weight of each token when detecting watermarks.

To comprehensively evaluate performance, we employ a suite of metrics for both output quality and watermark detection. For functional tasks like code generation and mathematical reasoning, we measure correctness using the pass@$k$ metric (Chen et al., 2021), calculating the proportion of $n > k$ samples that pass all hidden test cases, with a one-shot prompt for mathematical reasoning detailed in Appendix J.2. We assess generated comment quality against GPT-4o references (Appendix J.1) using word-level metrics METEOR, and the embedding-based BERTScore. Furthermore, we adopt the LLM-as-a-Judge paradigm with Stack-Eval (Shah et al., 2024; Zheng et al., 2023), using a powerful judge model to score outputs on a 0-3 scale for accuracy, completeness, and relevance (prompt in Appendix J.3); from this, we report the average score, the acceptance rate (scores $\geq 2$), and perplexity (PPL) for linguistic fluency. For watermark detection performance, we primarily use the Area Under the ROC curve (AUROC) as the main metric, and additionally report True Positive Rate (TPR) and F1-score under a False Positive Rate (FPR) constraint of less than 5%.

# 5 RESULTS

## 5.1 MAIN RESULTS

| Datasets | Metrics | Methods | | | | |
|---|---|---|---|---|---|---|
| | | KGW | SWEET-0.6 | SWEET-1.2 | EWD | CATMARK |
| **HUMANEVAL** | PASS@1 | 74.4 ±0.2 | 81.1 ±0.3 | **82.3** ±0.4 | 74.6 ±0.2 | **82.3** ±0.1 |
| | AUROC | 73.4 ±1.1 | 94.5 ±0.5 | 89.3 ±0.8 | 96.4 ±0.4 | **97.0** ±0.3 |
| | TPR | 21.3 ±1.5 | 67.7 ±1.2 | 43.9 ±1.3 | 81.7 ±0.9 | **82.9** ±0.9 |
| | METEOR | 23.9 ±0.1 | 24.1 ±0.1 | **25.5** ±0.2 | 23.9 ±0.1 | 24.2 ±0.1 |
| | BERTScore | 88.1 ±0.1 | 88.1 ±0.1 | **88.2** ±0.1 | 88.1 ±0.1 | 88.1 ±0.1 |
| **MBPP** | PASS@1 | 50.5 ±0.4 | 50.9 ±0.4 | 51.5 ±0.5 | 50.5 ±0.4 | **51.6** ±0.5 |
| | AUROC | 58.1 ±1.8 | 91.7 ±0.7 | 80.3 ±1.0 | 92.5 ±0.7 | **93.4** ±0.5 |
| | TPR | 10.4 ±2.0 | 65.8 ±1.5 | 33.4 ±1.8 | 64.4 ±1.6 | **67.2** ±1.2 |
| | METEOR | 10.6 ±0.2 | 10.9 ±0.2 | **11.4** ±0.3 | 10.6 ±0.2 | 11.1 ±0.2 |
| | BERTScore | 84.2 ±0.2 | 85.1 ±0.2 | 84.5 ±0.2 | 84.2 ±0.2 | **85.2** ±0.2 |
| **MATH-500** | PASS@1 | 68.6 ±0.6 | 70.0 ±0.5 | 69.4 ±0.5 | 68.6 ±0.6 | **71.6** ±0.4 |
| | AUROC | 85.0 ±0.4 | 99.5 ±0.1 | 94.3 ±0.5 | 99.8 ±0.1 | **99.8** ±0.1 |
| | TPR | 55.0 ±1.0 | 96.6 ±0.4 | 79.8 ±1.1 | 99.0 ±0.2 | **99.0** ±0.2 |
| **STACKEVAL** | AVG | 2.28 ±0.05 | 2.32 ±0.05 | 2.31 ±0.04 | 2.29 ±0.05 | **2.72** ±0.03 |
| | ACR | 90.8 ±0.8 | 92.4 ±0.7 | 92.4 ±0.7 | 91.2 ±0.8 | **97.5** ±0.3 |
| | PPL | 1.95 ±0.02 | 1.94 ±0.02 | **1.85** ±0.03 | 1.95 ±0.02 | 1.95 ±0.02 |
| | AUROC | 96.0 ±0.4 | 99.9 ±0.1 | 98.4 ±0.2 | 99.9 ±0.1 | **100.0** ±0.0 |
| | TPR | 85.2 ±1.0 | 99.4 ±0.2 | 93.0 ±0.6 | 99.8 ±0.1 | **100.0** ±0.0 |

Table 1: **Main results** of different cross-tasks performance and detection capability. For metrics in Stack-Eval, we use AVG to represent the average score and ACR to represent the acceptance rate. All methods use $\gamma = 0.5$ and $\delta = 2.0$. We vary the entropy threshold in SWEET (0.6 and 1.2) to present its impact on performance. For CATMARK, we set $\rho = 5$, $\alpha = -2$.

As demonstrated in Table 1, CATMARK achieves a superior synthesis of high-fidelity text generation and robust watermark detection, consistently outperforming baseline methods across a diverse set of cross-task benchmarks. In contrast, static threshold methods prove unable to adapt a single threshold to varied task demands. This inflexibility is evident with SWEET; on programming tasks, the SWEET-1.2 setting preserves better text quality than SWEET-0.6 but severely compromises watermark detection efficiency. However, for the Q&A-oriented StackEval task, this same SWEET-1.2 setting becomes broadly suboptimal, proving inferior to SWEET-0.6 in both judged quality and detection capability. Overcoming this fundamental limitation, CATMARK excels in both aspects concurrently. Our approach secures the highest or tied for highest pass@1 scores on the HumanEval, MBPP, and MATH-500 datasets and shows a substantial improvement in the LLM-as-a-Judge evaluation on StackEval with a leading average score and acceptance rate. This marked enhancement in generative quality is achieved without sacrificing security, as CATMARK also yields the highest watermark detection rates, registering top AUROC and TPR values across all tasks. In the Appendix D, we present the results using model Llama3.1-8B-InstructDubey et al. (2024).

## 5.2 EMPIRICAL ANALYSIS

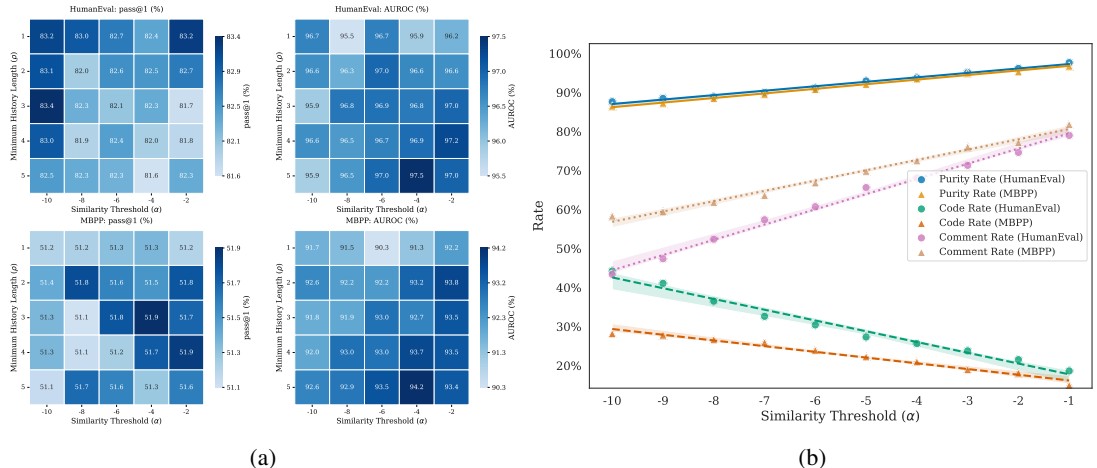

(a)                                   (b)

Figure 2: Hyperparameter sensitivity analysis for CATMARK with $\gamma = 0.5$ and $\delta = 2.0$ fixed. Sub-figure a displays performance stability on HumanEval and MBPP across similarity thresholds $\alpha \in \{-2, -4, -6, -8, -10\}$ and minimum entropy sequence lengths $\rho \in \{1, 2, 3, 4, 5\}$. Subfigure b illustrates the impact of $\alpha$ on the proportion of pure token categories with $\rho = 1$.

**Impact of Hyperparameters.** Figure 2 illustrates the impact of different hyperparameter combinations on the performance of CATMARK across various tasks.As shown in Figure 2a We employ two key parameters to constrain the watermark embedding: the similarity threshold $\alpha$, which is crucial for token classification, and the minimum entropy sequence length $\rho$, which assists in calculating the entropy threshold. To quantify stability, we compute the coefficient of variation ($C_v = \frac{\sigma}{\mu}$) for key metrics across the tested hyperparameter ranges. The $C_v$ for all metrics remained below 1%, with the largest fluctuation being a mere 0.96% for the AUROC on the MBPP dataset, confirming that CATMARK maintains stable performance in the different configurations of parameters and thus demonstrates the robustness of our proposed method. Furthermore, Figure 2b examines the effect of the similarity threshold $\alpha$ on token classification. As the value of $\alpha$ is increased, the proportion of tokens classified into pure categories rises, which is characterized by a decrease in the pure code category rate and a concurrent increase in the pure comment category rate. This trend suggests that comment tokens exhibit lower inter-token similarity compared to code tokens.

**Performance against Attack.** Attackers can remove watermarks from text through rewriting attacks before the watermarked text is detected, which causes detection performance drop. We remove watermarks using back-translation and paraphrase attacks, and evaluate the detection performance of our approach compared to baseline methods. Specifically, we first use the model to generate text on the MATH-500 task. In the back-translation attack, we translate the generated text into French and then back into English. In the para-phrase attack, we rewrite the generated text using a smaller Qwen2.5-7B-Instruct model. Figure 3a shows the changes in the ROC curves of different methods before and after the back-translation attack. Figure 3b shows the changes in the ROC curves of different methods before and after the paraphrase attack.

**Computational Overhead.** To evaluate the computational efficiency of our method, we conducted tim-ing experiments on HumanEval dataset. Our approach introduces additional computational steps during generation including KL divergence calculation for token categorization and dynamic entropy thresholding, which is also required for detection. As detailed in Appendix G, these mechanisms result in a marginal increase in generation latency. Our method achieves 33.1 tokens per second during generation, representing

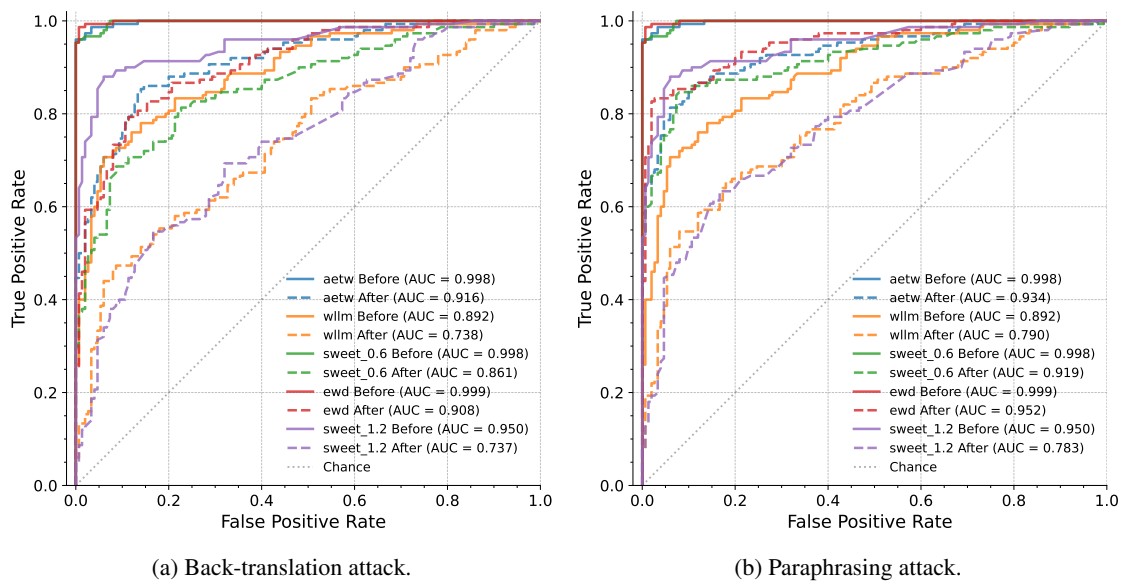

(a) Back-translation attack.

(b) Paraphrasing attack.

Figure 3: Watermark detection performance against two attacks. We set $\gamma = 0.5$ and $\delta = 2.0$ for watermark methods and $\rho = 5$, $\alpha = -2$ for CATMARK.

a 5.4% decrease compared to baseline approaches (SWEET: 35.1, WLLM: 36.8, EWD: 36.3 tokens per second). Notably, our detection latency remains highly competitive at 1.017 seconds per sample, outperforming EWD (1.031 seconds) despite its simpler methodology. These results demonstrate that the advanced dynamic capabilities of our algorithm are achieved with only a minimal and acceptable computational cost, confirming its practicality for real-world applications.

## 6 CONCLUSION

In this work, we introduced CATMARK, a dynamic framework designed to address the critical challenge of watermarking in cross-task scenarios where LLM-generated text contains heterogeneous content. By leveraging context-aware token categorization and adaptive entropy thresholding, CATMARK automates the watermarking process, eliminating the need for costly, task-specific calibration. This approach effectively balances the trade-off between detection robustness and text quality preservation. Our extensive experiments demonstrate that CATMARK significantly outperforms static-threshold baselines. It achieves state-of-the-art results by preserving high functional correctness while simultaneously ensuring superior detection robustness. The method's demonstrated adaptability to hybrid content, such as code with comments, highlights its practical utility for real-world LLM applications.

Furthermore, CATMARK exhibits strong resilience against common rewriting attacks, maintaining higher detectability after back-translation and paraphrasing compared to existing methods. However, we identify avenues for future improvement. The current framework, while effective, shows potential vulnerability to sophisticated redundancy injection attacks designed to artificially inflate entropy. Future work will focus on enhancing resilience to such adversarial manipulations and extending the context-aware framework to broader multimodal generation settings. By advancing adaptive watermarking strategies, this work paves the way for reliable provenance tracking of LLM outputs without compromising functional integrity, a critical step toward ethical AI deployment.

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

# A    USAGE OF LLM

After writing the paper, we used the LLM to polish and modify the grammar to make the expression of the paper more natural.

# B    PRELIMINARIES

This section introduces the foundational concepts necessary to understand our proposed Adaptive Entropy Threshold Watermarking (CATMARK) method. We will cover the text generation process of Large Language Models (LLMs) and the critical role of entropy in watermarking applications.

## B.1    LARGE LANGUAGE MODEL TEXT GENERATION

Large Language Models (LLMs) typically generate text in an auto-regressive manner. Given an input prompt $x = \{x_0, \ldots, x_{M-1}\}$ and a sequence of previously generated tokens $y_{<t} = \{y_0, \ldots, y_{t-1}\}$, the model predicts a probability distribution for the next token $y_t$. Specifically, at timestep $t$, the model outputs a logit vector $l_t \in \mathbb{R}^{|\mathcal{V}|}$ over the entire vocabulary $\mathcal{V}$. This vector is then converted into a probability distribution $p_t$ via the Softmax function:

$$p_{t,i} = \frac{e^{l_{t,i}}}{\sum_{j=1}^{|\mathcal{V}|} e^{l_{t,j}}} \tag{9}$$

where $p_{t,i}$ represents the probability of the $i$-th token in the vocabulary being the next token. Finally, the model samples the next token $y_t$ from this distribution $p_t$ using a decoding strategy such as multinomial sampling or beam search.

## B.2    SPIKE ENTROPY

To measure how spread out a distribution is, Kirchenbauer et al. (2023) proposed *spike entropy*. Given a discrete token probability vector $p$ and a scalar $m$, define the spike entropy of $p$ with modulus $m$ is:

$$S(p, m) = \sum \frac{p_k}{1 + m_k} \tag{10}$$

## B.3    THE CHALLENGE OF LOW-ENTROPY SCENARIOS

The performance of the KGW watermark is fundamentally linked to token entropy—a measure of the model's uncertainty in its prediction. We use Shannon Entropy for this measure:

$$H_t = -\sum_{k \in \mathcal{V}} p_{t,k} \log p_{t,k} \tag{11}$$

In high-entropy scenarios, the model's predictive distribution is flat, allowing the watermark bias $\delta$ to easily influence token selection. However, in low-entropy scenarios, such as code generation, the distribution is "spiky", with the model being highly confident about the next token. Modifying such a confident prediction can degrade text quality and functional correctness. Consequently, watermarked low-entropy text contains fewer green tokens, leading to low $z$-scores and detection failures.

---

**Algorithm 1** Watermark Generation in CATMARK

1: **Input:** Tokenized prompt $x = \{x_0, \ldots, x_{M-1}\}$, generated sequence $y_{[:t]}$, similarity threshold $\alpha$, minimum history $\rho$, green proportion $\gamma$, logit bias $\delta$
2: **Globals:** Categories $\mathcal{C} = \{(p_k, N_k, H_{h,k})\}_{k=1}^K$ per task, initially empty.
3: **for** step $t = M, M+1, \ldots$ **do**
4:    Compute logits $\mathbf{s}_t$ and entropy $H_t = -\sum_v P_t(v) \log P_t(v)$
5:    **for** each sequence in batch **do**
6:       Compute similarity $d_k = -\text{KL}(\sigma(\mathbf{s}_t) \parallel \sigma(\mathbf{p}_k))$ for all $k$
7:       $k^* \leftarrow \arg\max_k d_k$
8:       **if** $d_{k^*} \geq \alpha$ **then**
9:          Assign token to category $C_{k^*}$
10:          Update prototype: $\mathbf{p}_{k^*} \leftarrow \frac{N_{k^*}\mathbf{p}_{k^*} + \mathbf{s}_t}{N_{k^*}+1}$
11:          $N_{k^*} \leftarrow N_{k^*} + 1$
12:       **else**
13:          $K \leftarrow K + 1$
14:          Create $C_K$ with $\mathbf{p}_K \leftarrow \mathbf{s}_t$, $N_K \leftarrow 1$, and empty $H_{h,K}$
15:          $k^* \leftarrow K$
16:       **end if**
17:       Append $H_t$ to $H_{h,k^*}$
18:       Compute $\tau_{k^*}$ via Eq. 3
19:       **if** $H_t > \tau_{k^*}$ **then**
20:          Add $\delta$ to logits of green-listed tokens
21:       **end if**
22:    **end for**
23:    Sample $y_t$ from the modified distribution
24: **end for**

---

**Algorithm 2** Watermark Detection in CATMARK

**Input:** Token sequence $y = \{y_0, \ldots, y_{N-1}\}$, green token proportion $\gamma$, detection key.
**Output:** Detection result (positive if watermark is present).
**for** each token $y_t$ **do**
   Compute an entropy $H_t$ by Eq. 11.
   Update entropy sequence $H$.
**end for**
Compute a mean entropy $\mu_H$.
**for** each token $y_t$ with $H_t > \tau$ **do**
   Compute weight $W_t$ by Eq. 6.
**end for**
Apply KGW detection procedure to identify green token list $G$.
Compute weighted sum of green tokens $|s|_G$.
Compute z-score $z$ by Eq. 7.
**if** $z > $ predefined threshold **then**
   Return positive detection result.
**else**
   Return negative detection result.
**end if**

---

## C   WATERMARK ALGORITHM OF CATMARK

Algorithm 1 and Algorithm 2 demonstrate the process of applying and detecting watermarks in the CATMARK algorithm, where we use Shannon entropy to calculate the entropy value. Given a probability distribution vector p of a token, the entropy value of p can be calculated using Eq. 11.

## D   MORE RESULTS ON LLAMA

| Datasets | Metrics | Methods | | | | |
|---|---|---|---|---|---|---|
| | | KGW | SWEET-0.6 | SWEET-1.2 | EWD | CATMARK |
| **HUMANEVAL** | PASS@1 | 10.7 | 10.2 | 11.2 | 10.7 | **12.3** |
| | AUROC | 96.3 | 97.9 | 98.3 | **98.4** | **98.4** |
| | TPR | 85.4 | 87.2 | 92.7 | **93.9** | 90.2 |
| | METEOR | 20.5 | 20.5 | **22.9** | 20.5 | 21.4 |
| | BERTScore | 86.0 | 86.2 | **86.8** | 86.0 | 86.5 |
| **MBPP** | PASS@1 | 36.0 | 36.5 | **37.3** | 36.0 | 37.2 |
| | AUROC | 92.7 | 97.8 | 97.1 | **98.7** | **98.7** |
| | TPR | 65.4 | 89.2 | 88.6 | **94.2** | **94.2** |
| | METEOR | 11.5 | 11.4 | **12.3** | 11.5 | 11.6 |
| | BERTScore | 84.8 | 84.9 | **85.1** | 84.8 | 84.9 |
| **MATH-500** | PASS@1 | 25.6 | 25.2 | 26.2 | 25.6 | **27.8** |
| | AUROC | 95.4 | 99.7 | 99.8 | 99.3 | **99.9** |
| | TPR | 90.4 | 99.2 | 99.6 | 98.8 | **99.8** |
| **STACKEVAL** | AVG | 1.43 | 1.44 | 1.47 | 1.43 | **1.50** |
| | ACR | 49.6 | 52.2 | 53.1 | 49.6 | **54.6** |
| | PPL | 2.36 | 2.10 | **1.56** | 2.36 | **1.56** |
| | AUROC | 98.2 | **99.9** | 99.6 | 99.7 | **99.9** |
| | TPR | 95.8 | **99.4** | 99.1 | 99.3 | **99.4** |

Table 2: **LLama results** of different cross-tasks performance and detection capability. For metrics in StackEval, we use AVG to represent the average score and ACR to represent the acceptance rate. All methods use $\gamma = 0.5$ and $\delta = 2.0$. We vary the entropy threshold in SWEET (0.6 and 1.2) to present its impact on performance. Best results are bolded.

To verify the generalization capability of our method across different model architectures, we extended our evaluation to Llama-3.1-8B-Instruct, as detailed in Table 2. Consistent with the main results, CATMARK demonstrates a superior synthesis of high-fidelity text generation and robust watermark detection, proving its efficacy is model-agnostic. Static threshold methods like SWEET continue to exhibit a "trade-off" dilemma. For instance, SWEET-1.2 improves text metrics (e.g., METEOR) at the cost of detection efficiency (TPR). In contrast, CATMARK overcomes this limitation by achieving optimal performance in both dimensions simultaneously. Specifically, our method secures the highest Pass@1 scores across HumanEval, MBPP, and MATH-500, and achieves a leading average score of 1.50 with a 54.6% acceptance rate on StackEval.

# E    ABLATION STUDY

| Datasets | Metrics | Methods | | | |
|---|---|---|---|---|---|
| | | QBT | QBT+EWD | QBT+CAC | CATMark |
| **HUMANEVAL** | PASS@1 | 78.4 | 78.4 | **82.3** | **82.3** |
| | AUROC | 92.0 | 95.9 | 94.4 | **97.0** |
| | TPR | 68.9 | 80.5 | 73.2 | **82.9** |
| | METEOR | 24.0 | 24.0 | **24.2** | **24.2** |
| | BERTScore | **88.1** | **88.1** | **88.1** | **88.1** |
| **MBPP** | PASS@1 | 48.7 | 48.7 | **51.6** | **51.6** |
| | AUROC | **96.4** | **96.4** | 90.4 | 93.4 |
| | TPR | 85.0 | **85.4** | 60.0 | 67.2 |
| | METEOR | 11.0 | 11.0 | **11.1** | **11.1** |
| | BERTScore | 85.1 | 85.1 | **85.2** | **85.2** |
| **MATH-500** | PASS@1 | 46.0 | 46.0 | **71.6** | **71.6** |
| | AUROC | 99.3 | 99.4 | 99.5 | **99.8** |
| | TPR | 97.6 | 98.2 | 97.8 | **99.0** |
| **STACKEVAL** | AVG | 2.03 | 2.03 | **2.72** | **2.72** |
| | ACR | 71.1 | 71.1 | **97.5** | **97.5** |
| | PPL | **1.95** | **1.95** | **1.95** | **1.95** |
| | AUROC | 99.9 | **100.0** | 99.8 | **100.0** |
| | TPR | 99.5 | **100.0** | 99.0 | **100.0** |

Table 3: **Ablation Results.** We decouple CATMark to analyze the impact of different components. **QBT**: Quantile-Based Thresholding; **EWD**: Entropy-Weighted Detection; **CAC**: Context-Aware Clustering. Note that CATMark is equivalent to **QBT+CAC+EWD**. The best results are highlighted in **bold**.

To thoroughly decouple the individual contributions of our proposed framework, we conducted an ablation study isolating the effects of Context-Aware Clustering, Quantile-based Thresholding, and Entropy-Weighted Detection. Specifically, we evaluated three variants against the full CATMark method: (1) Quantile, which applies dynamic threshold without context differentiation; (2) Quantile + Entropy-Weighted, which incorporates weighted detection into the global quantile approach; and (3) Quantile + Clustering, which utilizes context-aware thresholds but employs standard unweighted detection. It is worth noting that we excluded the combination of clustering with a fixed threshold, as it is mathematically equivalent to a standard fixed-threshold approach when the threshold $\tau$ remains constant across all semantic states. The results, summarized in Table 3, provide two critical insights. First, the omission of the clustering mechanism (comparing *Quantile* vs. *Quantile + Clustering*) results in a notable degradation in generation quality, particularly in code generation tasks (Pass@1). This confirms that context-aware prototypes are essential for preserving utility in low-entropy scenarios. Second, removing the entropy-weighted detection module leads to a decline in detection performance (AUROC and TPR), validating its necessity for amplifying the watermark signal without compromising text fidelity.

# F    PERFORMANCE WITH DIFFERENT THRESHOLD FUNCTIONS

To assess the influence of the threshold function on our watermarking algorithm's efficacy, we compared four candidates, including functions with a decreasing characteristic: an exponential function ($e^{-x}$), a linear

| Functions | HumanEval | | | | MBPP | | | |
|---|---|---|---|---|---|---|---|---|
| | TPR(1%FPR) | TPR(5%FPR) | AUROC | pass@1 | TPR(1%FPR) | TPR(5%FPR) | AUROC | pass@1 |
| exp | 70.1 | **85.4** | **97.0** | **82.9** | **48.4** | **68.6** | **93.4** | 50.7 |
| linear | **71.9** | 85.4 | 96.7 | **82.9** | 42.6 | 63.4 | 92.1 | 50.3 |
| reciprocal | 0.0 | 0.0 | 50.0 | 81.4 | 0.0 | 0.0 | 49.7 | **51.8** |
| sigmoid | 59.1 | 84.1 | 96.6 | **82.9** | 12.0 | 67.2 | 92.8 | 50.1 |

Table 4: Comparison of code generation and detection performance metrics (pass@1, AUC, T(F < 5%)) across different function of CATMARK on HumanEval and MBPP datasets. We set $\gamma$= 0.5 and $\delta$=2.0 and addtionally $\rho = 5$, $\alpha = -2$.

| Metric | CATMARK | SWEET | KGW | EWD |
|---|---|---|---|---|
| Generation (s) | 16.801 | 16.256 | 15.361 | 15.557 |
| Detection (s) | 1.017 | 0.993 | 0.836 | 1.031 |
| Seconds/Token (gen) | 0.030 | 0.029 | 0.027 | 0.028 |
| Tokens/Second (gen) | 33.136 | 35.080 | 36.751 | 36.288 |

Table 5: This table shows the average time taken to generate more than 550 tokens texts using Qwen2.5-Coder-14B-Instruct on an NVIDIA RTX A800 80GB GPU, as well as the average time taken for detection measured in seconds

reciprocal ($x^{-1}$), a sigmoid function, and a baseline using the average entropy. The results in Table 4 reveal that while both decreasing functions aim to embed watermarks more selectively, their performance diverges significantly. The exponential function ($e^{-x}$) strikes the optimal balance, achieving an AUROC of 97.0 on HumanEval while preserving a high pass@1 score of 82.9. In contrast, the reciprocal function ($x^{-1}$), despite a similar design intention, fails completely (0.0% TPR), indicating that an overly aggressive reduction in watermarking opportunities undermines detectability. The linear and sigmoid functions show intermediate but less consistent results. This confirms that the specific nature of the decreasing function is critical, with $e^{-x}$ providing the most effective non-linear mapping for adaptive watermarking.

## G  COMPUTATIONAL OVERHEAD

During generation, CATMARKtakes 16.801 seconds on average—only 0.545 seconds (3.3%) slower than SWEET (16.256 s) and 1.440 seconds (9.4%) slower than the fastest baseline, KGW (15.361 s). This minor slowdown stems from the online clustering and per-cluster entropy thresholding steps, which require lightweight similarity computations and entropy tracking. Crucially, the per-token generation latency remains nearly identical across methods: CATMARKachieves 0.030 seconds/token (33.14 tokens/s), comparable to SWEET (0.029 s/token) and within 10% of KGW (0.027 s/token). This demonstrates that our context-aware watermarking does not bottleneck the autoregressive decoding loop.

For detection, CATMARKrequires 1.017 seconds—marginally slower than SWEET (0.993 s) but faster than EWD (1.031 s), and only 0.181 seconds (21.7%) slower than the most efficient detector, KGW (0.836 s). Given that detection is typically performed offline or in a verification pipeline (not in real-time generation), this sub-second latency is negligible for most applications.

## H    PROOF OF THEOREM 1

We begin our proof with a lemma from Kirchenbauer et al. (2023), which establishes a lower bound on the probability of sampling a token from the green list.

**Lemma H.1.** *Suppose a language model produces a raw probability vector $p \in (0,1)^{\mathcal{V}}$ over a vocabulary of size $|\mathcal{V}|$. The vocabulary is randomly partitioned into a green list $\mathcal{G}$ of size $\gamma|\mathcal{V}|$ and a red list of size $(1-\gamma)|\mathcal{V}|$. The logits for tokens in the green list are increased by a constant $\delta > 0$. If a token $k$ is sampled from this watermarked distribution, the probability that $k \in \mathcal{G}$ is lower-bounded by:*

$$\mathbb{P}[k \in \mathcal{G}] \geq \frac{\gamma e^{\delta}}{1 + (e^{\delta} - 1)\gamma} S_k(p, \frac{\gamma e^{\delta}}{1 + (e^{\delta} - 1)\gamma}) = \beta S_k$$

*where $S_k$ is the spike entropy of the token and we define $\beta = \frac{\gamma e^{\delta}}{1+(e^{\delta}-1)\gamma}$ for brevity.*

*Proof.* Let the generated token sequence be $y = \{y_0, \ldots, y_{N-1}\}$. The CATMARK detection method partitions the set of token indices $\mathcal{N} = \{0, \ldots, N-1\}$ based on an entropy threshold $\tau$ into a high-entropy set $\mathcal{I} = \{i \in \mathcal{N} \mid S_i > \tau\}$ and a low-entropy set $\mathcal{J} = \{i \in \mathcal{N} \mid S_i \leq \tau\}$.

The $z$-score statistic for a generic set of indices $\mathcal{M} \subseteq \mathcal{N}$ is given by:

$$z(\mathcal{M}) = \frac{\sum_{i \in \mathcal{M}} W_i \mathbb{1}_{i \in \mathcal{G}} - \gamma \sum_{i \in \mathcal{M}} W_i}{\sqrt{\gamma(1-\gamma) \sum_{i \in \mathcal{M}} W_i^2}}$$

where $W_i$ are token weights and $\mathbb{1}_{i \in \mathcal{G}}$ is the indicator function for the token being in the green list. The EWD method uses the full set $\mathcal{M} = \mathcal{I} \cup \mathcal{J}$, while CATMARK uses only the high-entropy set $\mathcal{M} = \mathcal{I}$.

Using Lemma H.1, we can establish a lower bound on the expected $z$-score by analyzing its numerator and denominator. The expected numerator for a set $\mathcal{M}$ is:

$$\mathbb{E}[\text{Num}(\mathcal{M})] = \sum_{i \in \mathcal{M}} W_i(\mathbb{P}[y_i \in \mathcal{G}] - \gamma) \geq \sum_{i \in \mathcal{M}} W_i(\beta S_i - \gamma)$$

Let's denote the lower bound on the signal from a set $\mathcal{M}$ as $L(\mathcal{M}) = \sum_{i \in \mathcal{M}} W_i(\beta S_i - \gamma)$. The condition in Theorem 1 establishes that for any token $y_j$ in the low-entropy set $\mathcal{J}$, the term $(\beta S_j - \gamma)$ is negative. Consequently, the total contribution from the low-entropy set to the signal's lower bound, $L(\mathcal{J})$, is also negative, then $L(\mathcal{I}) > 0$.

We now compare the $z$-score lower bounds for EWD and CATMARK.

$$z_{\text{EWD}} \geq \frac{L(\mathcal{I} \cup \mathcal{J})}{\sqrt{\gamma(1-\gamma) \sum_{i \in \mathcal{I} \cup \mathcal{J}} W_i^2}} \quad \text{and} \quad z_{\text{CATMARK}} \geq \frac{L(\mathcal{I})}{\sqrt{\gamma(1-\gamma) \sum_{i \in \mathcal{I}} W_i^2}}$$

For the denominator, we have $D(\mathcal{I} \cup \mathcal{J})^2 = D(\mathcal{I})^2 + D(\mathcal{J})^2$; since $D(\mathcal{J})^2 > 0$, the denominator for CATMARK is strictly smaller, $D(\mathcal{I}) < D(\mathcal{I} \cup \mathcal{J})$. These facts allow us to construct the following chain of inequalities:

$$Z_{\text{CATMARK}} = \frac{L(\mathcal{I})}{D(\mathcal{I})} > \frac{L(\mathcal{I})}{D(\mathcal{I} \cup \mathcal{J})} > \frac{L(\mathcal{I}) + L(\mathcal{J})}{D(\mathcal{I} \cup \mathcal{J})} = Z_{\text{EWD}}$$

$\square$

### H.1 EXPLANATION OF QUANTILE-BASED THRESHOLDING

Theorem 1 identifies a critical spike entropy boundary $S^\star = \gamma + (1 - \gamma)e^{-\delta}$. Tokens falling below this boundary contribute negatively to the detection statistic and should ideally be excluded. We define this *theoretical rejection region* for a cluster $k$ as:

$$A_k := \{i \mid S_i < S^\star\}. \tag{12}$$

However, the distribution of spike entropy, denoted by the CDF $F_k$, varies drastically across semantic contexts. A static scalar threshold is therefore suboptimal, as it ignores these distributional shifts. To robustly approximate $A_k$ without prior knowledge of $F_k$, CATMARK employs a data-driven thresholding strategy based on empirical quantiles.

**Empirical Formulation.** Let $\mathcal{H}_k = \{S_1, \ldots, S_{n_k}\}$ denote the historical spike entropy samples for cluster $k$, with the empirical cumulative distribution function (CDF) denoted by $\hat{F}_{n_k}$. The algorithm determines the exclusion threshold via the empirical quantile function $\hat{Q}_{n_k}(q)$, where the target quantile level $q$ is determined by a function of the cluster mean, $q_k = f(\mu_k)$.

The following theorem formally justifies the use of quantiles. It demonstrates that estimating the threshold via empirical quantiles provides a probabilistic guarantee that the algorithmic exclusion set covers the theoretical rejection region $A_k$, accounting for finite-sample noise.

**Theorem 2.** *Let $\tau_k := \hat{Q}_{n_k}(f(\mu_k))$ be the algorithmic threshold derived from the empirical quantile function. If the mapping function $f$ satisfies the conservatism condition:*

$$f(\mu_k) \geq F_k(S^\star) + \varepsilon \tag{13}$$

*for some precision $\varepsilon > 0$, then with probability at least $1 - 2e^{-2n_k\varepsilon^2}$, the algorithmic exclusion set $J_k := \{i \mid S_i \leq \tau_k\}$ satisfies:*

$$J_k \supseteq A_k. \tag{14}$$

*Proof.* This result leverages the Dvoretzky-Kiefer-Wolfowitz (DKW) inequality, which bounds the uniform convergence of the empirical CDF to the true CDF. For any $\varepsilon > 0$:

$$\mathbb{P}\left(\sup_x |\hat{F}_{n_k}(x) - F_k(x)| > \varepsilon\right) \leq 2e^{-2n_k\varepsilon^2}. \tag{15}$$

Consider the event where the empirical distribution is within the $\varepsilon$-bound. Let $q_k = f(\mu_k)$. The condition $f(\mu_k) \geq F_k(S^\star) + \varepsilon$ implies:

$$\hat{F}_{n_k}(S^\star) \leq F_k(S^\star) + \varepsilon \leq q_k. \tag{16}$$

By the definition of the empirical quantile function, the inequality $\hat{F}_{n_k}(S^\star) \leq q_k$ directly implies that the corresponding threshold satisfies $\tau_k = \hat{Q}_{n_k}(q_k) \geq S^\star$.

Consequently, for any token index $i \in A_k$ (where $S_i < S^\star$), it necessarily holds that $S_i < \tau_k$. Thus, $i \in J_k$, establishing the containment $A_k \subseteq J_k$. $\square$

## I CASE STUDY

To more intuitively illustrate the effectiveness of our watermarking algorithm, Figures 4, 5, 6, 7, 8, 9, 10, and 11 present a side-by-side comparison of watermark embedding and detection across different methods on the same task. The brown shaded regions, labeled as "ignored," represent the prompt. The black lines beneath each token indicate the weight assigned to that token during z-score computation. We use color intensity to represent the magnitude of these weights, where darker shades correspond to higher weights.

```python
def remove_Occ (string , letter ):
    # Remove first occurrence
    first_occ = string .index (letter )
    new_str = string [: first_occ ] + string [first_occ + 1 :]

    # Remove last occurrence if it exists
    if letter in new_str :
      last_occ = new_str .r index (letter )
      new_str = new_str [: last_occ ] + new_str [last_occ + 1 :]

    return new_str
```

Figure 4: KGW-MBPP

```python
def remove_Occ (String ): ch ):
    String =list (String )
    for i in range (len (String )):
        if ( String [i ]== ch ):
            String [i ]=' '
            break
    for i in range (len (String )- 1 ,- 1 ,- 1 ):
        if ( String [i ]== ch ):
            String [i ]=' '
            break
    String =' '. join (String )
    return String
```

Figure 5: SWEET-MBPP

```python
def remove_Occ (string , letter ):
    # Remove first occurrence
    first_occ = string .find (letter )
    if first_occ != - 1 :
      string = string [: first_occ ] + string [first_occ + 1 :]

    # Remove last occurrence
    last_occ = string .r find (letter )
    if last_occ != - 1 :
      string = string [: last_occ ] + string [last_occ + 1 :]

    return string
```

Figure 6: EWD-MBPP

```
def remove_O cc (String ,ch ):
    String = String .replace (ch ,' ', 1 )
    String = String [::- 1 ]
    String = String .replace (ch ,' ', 1 )
    String = String [::- 1 ]
    return  String
```

Green Token
Red Token
Ignored
Weight
Z-Score: 2.99

Figure 7: CAT-MBPP

```
    paren _string = paren _string .replace (' ', '')   # Remove spaces for
easier processing
    result = []
    current _group = []
    depth = 0

    for char in paren _string :
        current _group .append (char )
        if char == '(' :
            depth += 1
        elif char == ')' :
            depth -= 1

        # When depth returns to zero , we have a complete group
        if depth == 0 :
            result .append (' '. join (current _group ))
            current _group = []

    return result   # Return the list of separated groups """
```

Green Token
Red Token
Ignored
Z-Score: 2.50

Figure 8: KGW-HumanEval

```
    result = []
    stack = []
    current_group = []

    for char in paren_string.replace(" ", ""):    # Remove spaces and
iterate through characters
        if char == "(":    # Start of a new group or nested group
            stack.append(char)
            current_group.append(char)
        elif char == ")":    # End of a group or nested group
            stack.pop()
            current_group.append(char)
            if not stack:    # If stack is empty, we have a complete group
                result.append("".join(current_group))
                current_group = []    # Reset for the next group

    return result    # Return all separated groups
```

Legend: Green Token, Red Token, Ignored, Weight, Z-Score: 1.50

Figure 9: SWEET-HumanEval

```
    result = []    # List to store separated groups
    balance = 0      # Tracks balance of parentheses
    current_group = ''    # Current group being processed

    for char in paren_string.replace(' ', ''):    # Iterate over characters
, ignoring spaces
        current_group += char    # Add character to current group
        if char == '(':
            balance += 1    # Increase balance for an opening parenthesis
        elif char == ')':
            balance -= 1    # Decrease balance for a closing parenthesis

        # When balance is zero, a complete group is found
        if balance == 0:
            result.append(current_group)    # Add current group to result
            current_group = ''    # Reset current group for next iteration

    return result    # Return the list of separated groups
```

Legend: Green Token, Red Token, Ignored, Weight, Z-Score: 2.60

Figure 10: EWD-HumanEval

```
result = []    # List to store separated groups
stack = []     # Stack to track open parentheses
current_group = []    # List to build the current group of parentheses

for char in paren_string.replace(' ', ''):    # Iterate over characters
, ignoring spaces
    if char == '(':
        stack.append(char)    # Push open parenthesis onto stack
        current_group.append(char)    # Add to current group
    elif char == ')':
        stack.pop()    # Pop matching open parenthesis from stack
        current_group.append(char)    # Add to current group
        if not stack:    # If stack is empty, group is complete
            result.append(''.join(current_group))    # Join and add to
result
            current_group = []    # Reset for next group

return result    # Return the list of separated groups
```

Green Token
Red Token
Ignored
Weight
Z-Score: 2.65

Figure 11: CAT-HumanEval

**MBPP-Prompt**

**Prompt 1: Alphanumeric Check**

Write a function to check whether the given string is ending with only alphanumeric characters or not using regex.

Test Cases

```
assert check_alphanumeric("dawood@") == 'Discard'
assert check_alphanumeric("skdmsam326") == 'Accept'
assert check_alphanumeric("cooltricks@") == 'Discard'
```

```
import re
regex = '[a-zA-z0-9]$'
def check_alphanumeric(string):
    if(re.search(regex, string)):
        return ("Accept")
    else:
        return ("Discard")
```

**Prompt 2: Even Word Length Check**

Write a python function to check whether the length of the word is even or not.

Test Cases

```
assert word_len("program") == False
assert word_len("solution") == True
assert word_len("data") == True
```

```python
def word_len(s):
    s = s.split(' ')
    for word in s:
        if len(word)%2==0:
            return True
        else:
            return False
```

**Prompt 3: Find First Odd Number**

Write a python function to find the first odd number in a given list of numbers.

Test Cases

```
assert first_odd([1,3,5]) == 1
assert first_odd([2,4,1,3]) == 1
assert first_odd ([8,9,1]) == 9
```

```python
def first_odd(nums):
    first_odd = next((el for el in nums if el%2!=0),-1)
    return first_odd
```

**Prompt 4: Remove First and Last Occurrence**

Generate the function with comments after the docstring.
Requirements: - Only output the function (no docstring, test cases) with comments. - Place clear, concise English comments above each logical block of code (not inline). - Keep comments between 5–15 words. - Avoid redundancy or obvious descriptions. - Focus on explaining why something is done, not just what. - Do not generate any additional text after the code.
Write a python function to remove first and last occurrence of a given character from the string.

Test Cases

```
assert remove_Occ("hello","l") == "heo"
assert remove_Occ("abcda","a") == "bcd"
assert remove_Occ("PHP","P") == "H"
```

**HumanEval-Prompt**

**Prompt: Generate Function Body with Comments**

Generate the function body for the following function, adhering to the requirements listed below.

```python
from typing import List

def separate_paren_groups(paren_string: str) -> List[str]:
    """ Input to this function is a string containing multiple
    groups of nested parentheses. Your goal is to
    separate those group into separate strings and return the list
    of those.
    Separate groups are balanced (each open brace is properly closed
    ) and not nested within each other
    Ignore any spaces in the input string.
    """
```

**Requirements:**

- Only output the function body (no docstring, test cases) with comments.
- Place clear, concise English comments above each logical block of code (not inline).
- Keep comments between 5–15 words.
- Avoid redundancy or obvious descriptions.
- Focus on explaining why something is done, not just what.
- Do not generate any additional text after the code.



**Example from Docstring**

```
>>> separate_paren_groups('( ) (( )) (( )( ))')
['()', '(())', '(()())']
```



## J  DETAILED PROMPTS FOR EXPERIMENTS

### J.1  PROMPT FOR CODE COMMENTS GENERATION



**Prompt for Generating Reference Code Comments**

You are a **professional code reviewer**. Your task is to add clear, line-by-line English comments to the given Python function implementation.

**Each comment must:**

1. Explain what the line does (semantics)
2. Clarify why it's needed (intent)
3. Highlight any non-obvious logic or assumptions

**Guidelines:**

- Be concise and precise (5–15 words per comment)
- Use consistent style and terminology
- Avoid redundancy and obvious descriptions
- Follow PEP8 commenting conventions
- Place each comment on its own line *above* the corresponding code
- Prefix each comment with #
- Reflect the code accurately — no extra interpretation or added text

**Output Instructions:**

- Do **NOT** include any part of the original prompt in your output.
- **Only** return the solution with comments added, nothing else.

**For example:**



**Given Input**

```python
from typing import List

# <original_prompt>
def has_close_elements(numbers: List[float], threshold: float) ->
    bool:
    """ Check if in given list of numbers, are any two numbers
    closer to each other than
    given threshold.
    >>> has_close_elements([1.0, 2.0, 3.0], 0.5)
    False
    >>> has_close_elements([1.0, 2.8, 3.0, 4.0, 5.0, 2.0], 0.3)
    True
    """
# </original_prompt>

# <solution>
    for idx, elem in enumerate(numbers):
        for idx2, elem2 in enumerate(numbers):
            if idx != idx2:
                distance = abs(elem - elem2)
                if distance < threshold:
                    return True

    return False
# </solution>
```

After being added comment, you should **only** return:

**Expected Output**

```python
# Iterate over each element and its index in the list
for idx, elem in enumerate(numbers):
    # Iterate again to compare the current element with all others
    for idx2, elem2 in enumerate(numbers):
        # Ensure comparison is between different elements
        if idx != idx2:
            # Calculate the absolute difference between two elements
            distance = abs(elem - elem2)
            # Check if this difference is smaller than the threshold
            if distance < threshold:
                # Close pair found, return True immediately
                return True

# Return False if no elements are closer than the threshold
return False
```

---

**Prompt for Generating Commented Function Body**

Generate the function body of the following function with comments after the docstring: **Requirements:**

- Only output the function body (no function signature, docstring, or test cases).
- Place clear, concise English comments *above* each logical block of code (not inline).
- Keep comments between 5–15 words.
- Avoid redundancy or obvious descriptions.
- Focus on explaining **why** something is done, not just what.
- Do not generate any additional text after the code.

---

## J.2 PROMPT FOR MATH-500 REASONING

---

**Math Reasoning Prompt**

**Problem:**
What is the value of the expression $\frac{2025}{1+\frac{1}{1+\frac{1}{2025}}}$?

**Instructions:**
You are a helpful assistant that solves math problems step by step. Always conclude with the final answer in \boxed{}. Here's an example of how to solve a problem:

---

Example

---

**Problem:**
What is the area of the region defined by the equation $x^2 + y^2 - 7 = 4y - 14x + 3$?

**Solution:**
Let's think step by step.
We rewrite the equation as $x^2 + 14x + y^2 - 4y = 10$ and then complete the square, resulting in $(x + 7)^2 - 49 + (y - 2)^2 - 4 = 10$, or $(x + 7)^2 + (y - 2)^2 = 63$. This is the equation of a circle with center $(-7, 2)$ and radius $\sqrt{63}$. The area of this region is $\pi r^2 = 63\pi$. So the final answer is $\boxed{63\pi}$.

---

**Solution:**
Let's think step by step.

---

## J.3 PROMPT FOR STACKEVAL

---

**LLM-as-Judge Evaluation Prompt for StackEval**

You are a very experienced and knowledgeable answer checker. You will be given a question, a reference answer and an LLM generated answer. Your task is to evaluate how good the answer is

---

in answering the question of the user. More specifically, you will evaluate the acceptability of the answer for the user following the definition and rubric below.

**Acceptability Definition** Acceptability measures how effectively an answer satisfies the user's specific requirements and addresses their issue. It evaluates whether the response provides a viable solution, focusing on the answer's accuracy, relevance, and completeness. An acceptable answer is one that the user would regard as a fitting resolution to their query. An acceptable answer enables the user to proceed without requiring additional help or verification. An acceptable answer may not be perfect and may contain small inaccuracies that will not affect the usability of the provided answer. For example, if the answer is code, it must work without any user editing. If it is an advice, it must cover most crucial points.

**Acceptability Evaluation Rubric** Choose the most suitable category from the four-tiered scale provided to assess the acceptability of the response:

**Score 0 (Completely Unacceptable):**

- The answer is incorrect or entirely irrelevant, with substantial errors and no viable solution to the user's problem.
- Contains severe hallucinations or misinformation, significantly misleading the user.
- Leaves significant gaps, necessitating further search for information.
- The user would immediately disregard this answer and continue searching for a better solution.

**Score 1 (Useful but Unacceptable):**

- Contains some correct information but also significant inaccuracies or lacks important details, prompting additional research.
- Somewhat relevant but misses critical nuances, leading to an incomplete understanding.
- Not comprehensive, omitting important aspects and critical details needed to solve the user's problem.
- Provides some value but requires further searching for a complete and satisfactory solution.

**Score 2 (Acceptable):**

- Accurate, with correct information and guidance, free of critical errors that would prevent problem resolution.
- Relevant and demonstrates a clear understanding of the issue, addressing the main points and considerations, and directly applicable to the problem.
- Sufficiently complete, offering a satisfactory solution, even if it is not the most optimal solution, or a clear solution template that users can easily adapt. Minor details may be omitted, but nothing vital is missing.
- Provides enough information for the most user to proceed without additional help, even if some user-specific details need to be filled in. For example, it is ok if it has some examples URLs or templates to fill in with user data.

**Score 3 (Optimal):**

- The answer is 100% accurate and provides a detailed response, where the details improve answers quality and usability, with guidance that is specific and helpful for the user's particular issue.
- It is thorough, addressing not just the basic question but also touching on additional relevant aspects that could enhance the user's understanding of the solution.
- The response may include extra information, such as best practices or helpful tips, that adds value and could assist the user in avoiding common mistakes or in understanding the broader context.
- The user is likely to feel well-informed and be able to apply the solution effectively, with the answer being considered as reliable and optimal solution.

**Attention:** It is crucial to understand the threshold between Score 1 and Score 2: Score 1 is useful but unacceptable, where the answer provides some correct information but lacks completeness and desired accuracy, requiring the user to seek further information for most users, whereas Score 2 is acceptable, even if it is not perfect or optimal, offering accurate, relevant, and sufficiently complete information that allows the user to resolve their issue without needing additional resources.

**Assessment Guidelines**

1. Analyze the question and reference answer to pinpoint the core requirements for an acceptable answer to the user.

2. Carefully evaluate the generated answer for the given question by taking into account question's requirements and reference answer. The reference answer usually have solid points but they may not be the only way of solution.

3. Reason on the acceptability of the generated answer, analyze how acceptable the generated answer is. In the end of this reasoning, write 1 line of decision about its acceptability based on the definition and rubric above without a score.

4. Give your acceptability score based on all the observations above. Ensure your evaluation results are formatted into a valid JSON object.

**Output Format**   Ensure your evaluation results are formatted into a valid JSON object as outlined below:

```
{
    "questionAnalysis": "<str, Review the question to understand what
        core elements an LLM generated answer must include to satisfy the
        user>",
    "generatedAnswerAnalysis": "<str, Review the LLM-generated answer
        considering how good it covers the core elements in the
        questionAnalysis above, identifying both strengths and weaknesses.
         Highlight accurate, valuable aspects and pinpoint inaccuracies or
         irrelevant details.>",
    "acceptabilityEvaluation": "<str, Assess how well the generated
        answer meets the user's needs based on its accuracy, relevance,
        and completeness following the previous accuracy definition and
        rubric.>",
```

```
     "acceptabilityScore": "<int, Following the acceptabilityEvaluation,
        assign the most appropriate score from the acceptability rubric
        (0, 1, 2 or 3), be very accurate.>"
}
```

INPUTS

**User Question**
{{question}}

**Reference Answer**
{{answer}}

**LLM-Generated Answer**
{{completion}}

