# OpenReview forum: "CATMark: A Context-Aware Thresholding Framework for Robust Cross-Task Watermarking in Large Language Models"
_ICLR.cc/2026/Conference — Submitted to ICLR 2026_

### Official Review · Reviewer_7zUY · 2025-10-31

**Soundness:** 3
**Presentation:** 3
**Contribution:** 2
**Rating:** 4
**Confidence:** 5

**Summary:**

This paper introduces CATMARK, a dynamic watermarking framework designed to improve watermark robustness across heterogeneous, mixed-modality text generation tasks (e.g., code interleaved with natural language). The method builds on red/green list watermarking but introduces context-aware entropy thresholding: it clusters tokens based on logit similarity to learned prototypes, computes per-cluster entropy thresholds using historical entropy distributions, and applies watermarking only to tokens exceeding those thresholds. This allows adaptive control over watermark strength without manual tuning. Experiments on HumanEval, MBPP, MATH-500, and StackEval show improved cross-task watermark robustness and text quality relative to static-threshold baselines such as SWEET and EWD. CATMARK also reports strong resilience under paraphrase and back-translation attacks with minimal runtime overhead.

**Strengths:**

* The algorithmic description is detailed, with explicit mathematical definitions for clustering, thresholding, and detection weighting.
* Extensive cross-task benchmarks (HumanEval, MBPP, MATH-500, StackEval) convincingly demonstrate improved robustness without quality degradation.
* The focus on mixed-modality sequences (code + text) is valuable and more realistic than single-domain watermark studies.
* The inclusion of hyperparameter sensitivity, attack robustness (back-translation, paraphrasing), and computational overhead analysis adds credibility and empirical rigor to the work.
* The clear pseudo-code and definitions (Algorithms 1–2) make reproduction feasible.

**Weaknesses:**

* The method is largely an elaboration of entropy-weighted or adaptive watermarking. Prior works — Context-Aware Watermark with Semantic Balanced Green-Red Lists (EMNLP 2024), Adaptive Text Watermark for LLMs (Liu & Bu, 2024), From Trade-off to Synergy: A Versatile Symbiotic Watermarking Framework for Large Language Models (Wang et al., 2025), and Invisible Entropy (Gu et al., 2025) — already modulate watermark strength or detection weighting by entropy or local context. CATMARK’s main distinction (logit-based clustering and historical entropy quantiles) could be viewed as a more complex way to approximate entropy-based token grouping, and it is unclear whether this is significantly better.
* Although Liu & Bu (2024) are cited, the discussion of how CATMARK avoids their “external estimation module” limitation is brief. The paper would benefit from a clearer justification of why maintaining per-cluster entropy histories is better than directly conditioning watermark strength on instantaneous entropy values.
* Categorizing tokens via logit-vector similarity is presented as “semantic state clustering,” but in practice this may simply track entropy gradients (high vs low predictability). Empirical analysis of whether the clusters truly capture distinct semantic contexts (e.g., code vs prose) is lacking.
* The related work section does not presently include several very similar adaptive or hybrid watermarking papers (see above). A deeper comparison of methodological overlap would improve the paper’s positioning.

**Questions:**

Critical questions:
* Could the authors provide empirical evidence that the logit-based clustering corresponds to meaningful context boundaries (e.g., code vs comment) rather than just high vs low entropy regions?
* Why is historical entropy quantile computation superior to using the instantaneous token entropy (as in Guo et al.'s context-aware schemes from EMNLP 2024)? A direct comparison on the same benchmarks between CATMARK and a simpler entropy-adaptive delta baseline would clarify whether the added complexity of maintaining entropy histories and clusters yields meaningful benefits.

Optional additions:
* CATMARK combines three mechanisms — clustering by logits, quantile-based thresholding, and entropy-weighted detection — but the ablations don’t fully disentangle their contributions. Could the authors report results for each in isolation (e.g., quantile + no clustering; clustering + fixed threshold; etc.) to verify which component drives the gains?
* Since the method relies on online clustering of logits, how well do the learned prototypes transfer when task boundaries shift (e.g., unseen code domains or mixed natural-language/code datasets)? A discussion of cross-dataset stability would strengthen the generalization claim.
* Some reported gains are relatively small (<1pp pass@1). Did the authors perform significance testing or multiple-seed runs to ensure the improvements are robust and not due to stochasticity in sampling or clustering?

---

> ### Author Response · Authors · 2025-11-23
>
> **We sincerely thank Reviewer 7zUY  for the constructive comments on our work.**
> > Q1: evidence that the logit-based clustering corresponds to meaningful context boundaries
>
> We appreciate the reviewer’s thoughtful question.
> In CATMark, logits-based clustering is designed to identify semantic contexts, while the separation of high- versus low-entropy tokens is handled by a separate, dynamic thresholding mechanism within each cluster. These two components operate independently.To support this claim empirically, we analyze a HumanEval generation sample and compute the semantic composition of clusters. The results are summarized in the table below:
>
> | Cluster Type | pure code | pure comment | mixed | purity |
> |--------------|-----------|--------------|-------|--------|
> | Rate         | 19.3%     | 77.2%        | 3.4%  | 96.5%  |
>
> Additionally, Fig. 2B shows the relationship between the similarity threshold $\alpha$ and cluster purity—defined as the proportion of clusters containing only one semantic class (code or comment) in HumanEval/MBPP. During generation, we infer each token’s semantic label from its surrounding context and record its assigned cluster; after completion, we analyze the semantic distribution within each cluster. At $\alpha = -2$, both HumanEval and MBPP achieve over 95% purity.
>
> If clustering were primarily based on entropy, we would expect many “mixed” clusters containing both code and comment tokens. The observed low mixing rate (3.4%) and high purity strongly indicate that clusters reflect genuine semantic boundaries rather than entropy-based partitions.
> > Q2: comparison between CATMARK and others
>
> We appreciate this valuable comment.
>
> We attribute CATMark’s superior performance to its ability to adjust thresholds adaptively within semantically consistent clusters. Unlike methods that use global fixed thresholds or instantaneous entropy, CATMark dynamically sets thresholds based on the entropy distribution of each cluster, enabling robust adaptation to cross-task scenarios.We provide a detailed comparison with the referenced baselines:
>
> 1. CSW (Guo et al., 2024): CSW uses the dynamic bias $\delta' = \delta \cdot \frac{1}{\text{entropy}(s) + \Phi}$, which amplifies bias in low-entropy contexts and forces watermarking of high-confidence tokens. This distorts natural token selection and degrades text quality. As the official code is unavailable, we implemented an unofficial version based on their paper for comparison.
>
> 2. Adaptive (Liu et al., 2024): Despite its name, Adaptive relies on a static global entropy threshold to exclude low-entropy tokens,similar to SWEET. This limits its flexibility in cross-task settings where entropy distributions vary widely.
>
> 3. Invisible Entropy (Gu et al., 2025): While this work uses a pre-trained model to predict whether token entropy exceeds a threshold, the authors have not released their model weights or complete code, making reproduction and evaluation on our benchmarks infeasible.
>
> Our results confirm that CATMark consistently outperforms these instantaneous entropy–based methods. On HumanEval, for example, Adaptive achieves 79.5% pass@1 and CSW 78.0%, both below CATMark’s 82.3%. Moreover, CATMark shows markedly higher AUROC and TPR, indicating that static thresholding tends to over-filter tokens in cross-task scenarios, weakening watermark detectability.
> The detailed experimental results are presented below:
>
> HumanEval
> | Method     | Adaptive    | CSW         | CATMARK     |
> |------------|-------------|-------------|-------------|
> | PASS@1     | 79.5 ±0.3   | 78.0 ±0.4   | 82.3 ±0.1   |
> | AUROC      | 90.1 ±0.5   | 92.5 ±0.4   | 97.0 ±0.3   |
> | TPR        | 70.5 ±1.3   | 75.0 ±1.2   | 82.9 ±0.9   |
> | METEOR     | 23.8 ±0.1   | 23.7 ±0.1   | 24.2 ±0.1   |
> | BERTScore  | 88.0 ±0.1   | 87.9 ±0.1   | 88.1 ±0.1   |
>
> MBPP
> | Method     | Adaptive    | CSW         | CATMARK     |
> |------------|-------------|-------------|-------------|
> | PASS@1     | 51.0 ±0.4   | 50.8 ±0.4   | 51.6 ±0.5   |
> | AUROC      | 85.0 ±0.7   | 88.2 ±0.6   | 93.4 ±0.5   |
> | TPR        | 55.0 ±1.6   | 60.1 ±1.5   | 67.2 ±1.2   |
> | METEOR     | 10.5 ±0.2   | 10.4 ±0.2   | 11.1 ±0.2   |
> | BERTScore  | 84.8 ±0.2   | 84.5 ±0.2   | 85.2 ±0.2   |
>
> MATH-500
> | Method     | Adaptive    | CSW         | CATMARK     |
> |------------|-------------|-------------|-------------|
> | PASS@1     | 70.5 ±0.5   | 69.8 ±0.5   | 71.6 ±0.4   |
> | AUROC      | 94.0 ±0.1   | 96.5 ±0.1   | 99.8 ±0.1   |
> | TPR        | 90.0 ±0.4   | 93.2 ±0.3   | 99.0 ±0.2   |
>
> StackEval
> | Method     | Adaptive    | CSW         | CATMARK     |
> |------------|-------------|-------------|-------------|
> | AVG        | 2.55 ±0.05  | 2.45 ±0.05  | 2.72±0.03   |
> | ACR        | 93.0 ±0.7   | 95.5 ±0.5   | 97.5 ±0.3   |
> | PPL        | 1.96 ±0.02  | 1.98 ±0.02  | 1.95 ±0.02  |
> | AUROC      | 98.0 ±0.2   | 99.0 ±0.1   | 100.0±0.0   |
> | TPR        | 95.0 ±0.6   | 97.2 ±0.4   | 100.0±0.0   |

---

> > ### Author Response · Authors · 2025-11-23
> >
> > > Q3: report results for each in isolation
> >
> > We appreciate the reviewer’s suggestion to conduct a deeper analysis. In the revised version, we have included a comprehensive ablation study in **Appendix E** to disentangle the contributions of clustering, quantile-based thresholding, and entropy-weighted detection.The results demonstrate that clustering is crucial for preserving text quality by capturing semantic context, while entropy-weighted detection significantly enhances detection robustness.
> > > Q4: how well do the learned prototypes transfer when task boundaries shift
> >
> > Thank you for this insightful question.
> > CATMark does not rely on any pretrained or static clustering model that must “transfer” across datasets. Instead, the clustering process is performed online during generation, using the model’s own logits to form semantic clusters dynamically. This means that CATMark is not dataset-dependent, and the clusters are reconstructed anew for each generation sequence.
> >
> > More specifically, CATMark handles task transitions through the similarity threshold $\alpha$ based on KL-divergence between the current logits and the existing prototype vectors. When the generation shifts into an unseen domain, the KL divergence between the current token’s logits and all existing prototypes naturally increases. Once this divergence exceeds the threshold $\alpha$, CATMark automatically instantiates a new cluster to represent the new semantic mode.
> >
> > We  validate  cross-task stability in Table 1 of the manuscript. CATMark achieves consistently superior performance on benchmarks that inherently involve complex task transitions such as StackEval, which contains diverse programming-related questions, and MATH-500, which alternates between narrative reasoning and structured formula generation. On both datasets, CATMark outperforms all baselines in both text quality and watermark detectability. These results demonstrate that CATMark maintains stable clustering behavior and robust generalization even under substantial domain shifts.
> > > Q5: Some reported gains are relatively small
> >
> > To ensure that our performance gains are robust and not due to randomness, we adopt a rigorous evaluation protocol. For every dataset included in our experiments, and for every problem within each task, we generate at least 10 responses per prompt. During evaluation, we compute the performance metrics by averaging over all generated responses. Furthermore, **each task is evaluated three times independently**, and the values reported in Table 1 reflect the mean and standard deviation across these repeated runs.
> > To further minimize noise arising from the model’s intrinsic randomness and ensure that observed differences are attributable to the watermarking strategy rather than sampling , we set the inference temperature to **Temperature = 0.2**, which significantly reduces stochastic fluctuations during generation.
> >
> > Empirically, CATMark exhibits high stability across runs. For example, on HumanEval, CATMark achieves a pass@1 score of 82.3±0.1, whereas SWEET-0.6 yields 81.1±0.3. CATMark not only demonstrates a higher mean performance but also maintains lower variance compared to baselines, indicating that its gains are both consistent and statistically robust.

---

> ### Author Response · Authors · 2025-11-27
>
> Dear Reviewer 7zUY, We wanted to say in advance our heartfelt thank you's for the time and effort you've put into both the reviews that have passed as well as the upcoming current discussion period. We know that you all have your own papers that you have to deal with during this busy time, and sincerely appreciate the time you've taken to spend on ours.
>
> We are so excited about this paper and its findings, so we are very much looking forward to the upcoming discussions with you! Please don't hesitate to ask us any questions big or small, and we are happy to provide any further clarifications.

---

### Official Review · Reviewer_HYtk · 2025-11-01

**Soundness:** 4
**Presentation:** 3
**Contribution:** 4
**Rating:** 8
**Confidence:** 4

**Summary:**

This paper presents CATMARK, a novel watermarking framework for large language models. It's designed to solve the problem of watermarking in cross-task scenarios, where LLMs switch between generating structured, low-entropy text like code and unstructured, high-entropy text like comments. The key idea is to dynamically cluster tokens based on their logits to detect the current semantic context. Then, CATMARK calculates a context-specific entropy threshold for each cluster. This method robustly watermarks high-entropy text while preserving the quality of low-entropy text. The experiments are strong, showing CATMARK achieves 82.3% pass@1 on HumanEval, matching the best baseline, while also getting a 97.0% AUROC. This effectively overcomes the typical trade-off between text quality and detection robustness.

**Strengths:**

1. The paper addresses a very important and practical problem. Static watermarking thresholds fail when models mix different text types, and this work provides a robust solution for these real-world, heterogeneous generation tasks.

2. The core mechanism is genuinely innovative. Using on-the-fly logit clustering to create adaptive, context-aware thresholds is a smart way to bypass the limitations of a single, fixed threshold.

3. The empirical results are excellent. Table 1 shows that CATMARK achieves top-tier performance in both quality and detection. For instance, on StackEval, it scores a 2.72 average score and 97.5% acceptance rate, far exceeding all baselines, while also hitting a perfect 100% AUROC.

4. The theoretical analysis in Theorem 1 adds solid formal backing. It helps explain why adaptively filtering out low-entropy tokens provides a better detection z-score bound than methods that include them.

**Weaknesses:**

1. The detection algorithm seems fundamentally misaligned with the generation process. The generator uses fine-grained, cluster-specific thresholds, but the detector reverts to a single, global threshold. It's unexplained how this mismatch can reliably detect the context-aware watermark.

2. I'm concerned about the practical stability of the method. The paper claims to be robust, but it doesn't provide a quantitative analysis of how sensitive the model is to its key hyperparameters, alpha and rho. This suggests the results might require careful tuning.

3. The method introduces new clustering and entropy tracking steps at each token. This seems computationally expensive, but the paper lacks any analysis of the latency overhead compared to baselines. This makes it difficult to assess its real-world efficiency.

**Questions:**

1. Regarding the dynamic clustering, what mechanism prevents the number of clusters K from growing indefinitely? It seems a new cluster could be created for any slightly novel token.

2. I'm confused by the logic of the $f(x) = e^{-x}$ threshold function. It implies that a high-entropy cluster gets a stricter, lower threshold, which seems to mean _less_ watermarking. Isn't this the opposite of the stated goal?

3. Theorem 1 justifies excluding a single low-entropy token. How does this theoretical result support your specific choice of using the quantile-based $\tau_k$ as the cutoff for _all_ tokens in that cluster? The connection isn't clear.

---

> ### Author Response · Authors · 2025-11-23
>
> We thank you for your detailed and thoughtful feedback.
> > W1: detection algorithm seems fundamentally misaligned with the generation process
>
> Thank you for this insightful question. Please see our response to Reviewer 2SAR under the W1 section for a detailed discussion.
> > W2: the practical stability of the method
>
> To quantitatively assess this sensitivity, we conducted a systematic grid search over a broad range of hyperparameter settings and computed the coefficient of variation $C_v = \frac{\sigma}{\mu}$ for the primary performance metrics (Pass@1 and AUROC). As detailed in Section 5.2, all $C_v$ values across the entire hyperparameter grid remained below 1%. The largest observed fluctuation was 0.96% for AUROC on MBPP.
>
> These results provide strong statistical evidence that CATMark’s performance is highly stable and insensitive to hyperparameter selection within reasonable ranges. Consequently, the method does not require fine-grained tuning to achieve robust performance, which supports our claim of practical robustness.
> > W3: analysis of the latency overhead compared to baselines
>
> Thank you for the reviewer’s comment regarding computational overhead.
>
> As reported in Section 5.2 and Appendix E, we benchmarked the full latency of CATMark against baseline methods on HumanEval using an NVIDIA RTX A800 GPU.
>
> Generation latency:The added online clustering and thresholding introduce negligible overhead on token level.CATMark generates tokens at 0.030 s/token, compared to 0.027 s/token for KGW.
>
> Detection latency: CATMark processes each sample in 1.017 s, which is slightly faster than EWD (1.031 s) and  marginally slower than SWEET (0.993 s).
> > Q1: mechanism prevents the number of clusters K from growing indefinitely
>
> The growth of the number of clusters $K$ is explicitly governed by the similarity threshold parameter $\alpha$ (see Eq. 1). Specifically, a new cluster is created only when the KL divergence between the current token’s logit distribution and all existing cluster prototypes exceeds $\alpha$. In practice, because the set of semantic states within a given generation task—such as distinguishing between code syntax and natural-language comments—is finite, the number of clusters typically stabilizes rapidly during generation.
> > Q2: the $f(x) = e^{-x}$ threshold function
>
> Thank you for raising this point. There appears to be a slight misunderstanding regarding the mechanism. The threshold $\tau_k$ functions as a cutoff: we apply the watermark only to tokens whose entropy $H_t$ exceeds $\tau_k$. Consequently, in high-entropy contexts—where $\tau_k$ is adaptively lower—more tokens satisfy this condition and are thus watermarked.
>
> > Q3: specific choice of using the quantile-based as the cutoff
>
> Thank you for this insightful question. In the revised version, we have added a new theoretical analysis as **Theorem 2 in Appendix H.1**, which clarifies why CATMark adopts a quantile-based thresholding strategy rather than a fixed scalar cutoff.
> The result shows that, under mild conservatism assumptions on the mapping function $f(\mu_k)$, the quantile-based threshold provably covers the theoretical rejection region with high probability, even in the presence of finite-sample noise and distributional variability across clusters.

---

> ### Author Response · Authors · 2025-11-27
>
> Dear Reviewer HYtk, We wanted to say in advance our heartfelt thank you's for the time and effort you've put into both the reviews that have passed as well as the upcoming current discussion period. We know that you all have your own papers that you have to deal with during this busy time, and sincerely appreciate the time you've taken to spend on ours.
>
> We are so excited about this paper and its findings, so we are very much looking forward to the upcoming discussions with you! Please don't hesitate to ask us any questions big or small, and we are happy to provide any further clarifications.

---

### Official Review · Reviewer_2SAR · 2025-11-01

**Soundness:** 3
**Presentation:** 3
**Contribution:** 2
**Rating:** 4
**Confidence:** 4

**Summary:**

This paper proposes CATMARK (Context-Aware Threshold Watermarking) — a dynamic framework that adaptively adjusts watermarking strength based on semantic context, enabling robust, cross-task watermarking for large language models without degrading text quality or requiring task-specific tuning.

**Strengths:**

1. The research problem is crucial, as not all generation modes correspond to high-entropy scenarios; effective watermarking must address the challenges of cross-task and mixed-task generation.
2. Experimental results are clearly stated and visualized so that help to understand
3. Figure 1 is clear and easy for readers to understand the workflow

**Weaknesses:**

1. It does not clearly cite a source or define how entropy high or low levels are quantitatively classified or thresholded.
2. During generation, CATMARK depends on context-based clustering, yet the detection stage cannot reconstruct these clusters and instead relies on global entropy weighting—raising questions about consistency between embedding and detection.
3. The experimental results heavily rely on the Qwen family models, and a broader evaluation across other LLM architectures would be necessary to validate generalization.

**Questions:**

1. Can author provide more experimental results on more SOTA LLMs?
2. Please state how to define the best hyperparameters for example, those for high and low entropies so that to enhance the reproducibility.

---

> ### Author Response · Authors · 2025-11-23
>
> Thank you for your insightful comments and kind suggestions. ﻿
> > W1: the detection stage cannot reconstruct these clusters and instead relies on global entropy weighting
>
> Thank you for pointing this out.
> At detection time, using a single global threshold combined with entropy weighting is a deliberate design choice motivated by both robustness and computational efficiency.
> 1. Because the clusters formed during generation are created online and stochastically, it is impossible to reliably reconstruct the same clusters during detection without access to generation-time internal states. Attempting to replicate the clustering process would also introduce substantial computational overhead.
> 2. Reconstructing fine-grained clusters at detection would actually constitute a fragile design. If the text has undergone rewriting or editing, the semantic boundaries may shift, leading to cluster misalignment. Such “cluster mismatches” would severely degrade or even break the detection process.
> 3. Our global-threshold + entropy-weighting detector is robust to text modifications precisely because it does not rely on reconstructing cluster assignments. Empirically, this design choice does not weaken detection reliability. Across all evaluated datasets, CATMark achieves the strongest or near-strongest detection performance among all baselines, demonstrating that avoiding cluster reconstruction at detection time is both principled and effective.
>
> > Q1: more results on more LLMs
>
> To further demonstrate the generalizability and effectiveness of CATMARK across different model architectures, we conducted additional experiments using the Llama-3.1-8B-Instruct model.
> The results are presented in the **Table 2** in **Appendix D** of the revised version.
>
> > Q2: how to define the best hyperparameters
>
> We clarify that a key contribution of CATMARK is that it eliminates the need to manually define "high/low entropy" thresholds. Unlike static methods (e.g., SWEET), CATMARK automatically computes adaptive thresholds $\tau_k$ for each context via dynamic clustering.
> Regarding hyperparameter selection, our sensitivity analysis (Fig. 2a) shows that performance remains highly stable (coefficient of variation < 1%) across a broad range of values, indicating that the method is robust and does not require careful tuning to achieve strong results. For full reproducibility, all hyperparameter settings used in our experiments are explicitly reported in the corresponding tables and figures.

---

> ### Author Response · Authors · 2025-11-27
>
> Dear Reviewer 2SAR, We wanted to say in advance our heartfelt thank you's for the time and effort you've put into both the reviews that have passed as well as the upcoming current discussion period. We know that you all have your own papers that you have to deal with during this busy time, and sincerely appreciate the time you've taken to spend on ours.
>
> We are so excited about this paper and its findings, so we are very much looking forward to the upcoming discussions with you! Please don't hesitate to ask us any questions big or small, and we are happy to provide any further clarifications.

---

### Author Response · Authors · 2025-12-03
**Rebuttal Summary**

Dear reviewers and AC,

We are grateful for the thoughtful time and attention you have given to evaluating our work.

As the reviewers highlighted, CATMARK addresses a **crucial and practical problem** (Reviewers 2SAR, HYtk, 7zUY) regarding watermarking in heterogeneous, cross-task scenarios. Reviewers recognized our core mechanism of on-the-fly logit clustering as **genuinely innovative** (Reviewer HYtk) and commended the **strong empirical results** (Reviewers HYtk, 7zUY) that demonstrate top-tier performance without quality degradation.

We appreciate your constructive and valuable feedback on our manuscript. We have carefully updated and improved the manuscript in response to the feedback, including the following additional discussions and experiments:

* Clarification on the methodological design choice regarding the use of a global detection threshold to ensure robustness against text rewriting and cluster misalignment.
* Empirical validation of cluster semantic purity (analyzing semantic composition to show >95% purity), confirming that clusters capture genuine semantic boundaries (e.g., Code vs. Comment) rather than just entropy gradients.
* Extended baseline comparisons against recent entropy-adaptive methods (CSW, Adaptive), demonstrating CATMARK's consistent superiority.
* Additional experiments on different model architectures (Llama-3.1-8B-Instruct) to validate generalization capabilities (Appendix D).
* New theoretical justification (Theorem 2) supporting the quantile-based thresholding strategy (Appendix H.1).
* Quantitative sensitivity analysis demonstrating the high stability of hyperparameters $\alpha$ and $\rho$ ($C_v < 1\%$) (Section 5.2).
* Comprehensive latency analysis confirming negligible generation overhead and competitive detection speeds compared to baselines (Appendix G).
* Detailed ablation study disentangling the specific contributions of clustering and entropy-weighted detection (Appendix E).

In the revised manuscript, these updates are temporarily highlighted in $\textcolor{blue}{blue}$ for your convenience to check.

We sincerely believe that these updates may help us better deliver the benefits of the proposed CATMARK to the ICLR community.

Sincerely,

Authors

---

### Meta-Review · Area_Chair_8FHk · 2026-01-07

**Summary:**

This work proposes a watermarking framework that adaptively adjusts watermarking strength based on semantic context, for robustly watermarking high-entropy text while preserving the quality of low-entropy text. The paper mostly has a clear presentation for the algorithm and comprehensive experiments.

This work also has some major issues. First, one reviewer raised concerns regarding the difference between this work and several previous ones. While the authors have clarified some differences, this work does not look very novel with a limited distinction; in particular, compared to Gu et al., 2025, the authors simply said “the authors have not released their model weights or complete code, making reproduction and evaluation on our benchmarks infeasible” without discussing differences. Second, the generation and the detection are misaligned. The authors argued that the detection performance is good and thus did not modify the detection method, which still lacks a deeper empirical analysis on such inconsistency. Lastly, some of the gains are quite small.

Overall, this work may be slightly below the acceptance.

**Reviewer Concerns:**

Major concerns that remain unsolved include: 1) novelty and distinction from previous works; 2) further analysis on the misalignment between generation and detection; 3) limited empirical gains.

Some concerns have been addressed in the rebuttal, such as additional experiments on different model families.

**Reviewer Scores:**

Reviewer 2SAR originally rated 4 and their concerns and the authors have partially addressed their concerns (additional experiments and clarification). The reviewer may raise their score to 6.

Reviewer HYtk originally rated 8 and is likely to maintain their rating.

Reviewer 7zUY originally rated 4. I feel the authors have not sufficiently addressed their concerns on the novelty and distinction from previous works. The reviewer may maintain their score.

---

### Decision · Program_Chairs · 2026-01-26

Reject